



# Source-receptor matrix calculation for deposited mass with the Lagrangian particle dispersion model FLEXPART v10.2 in backward mode

Sabine Eckhardt[1], Massimo Cassiani[1], Nikolaos Evangeliou[1], Espen Sollum[1], Ignacio Pisso[1] and Andreas Stohl[1]

[1]NILU - Norwegian Institute for Air Research, Kjeller, Norway

*Correspondence to*: Sabine Eckhardt (sabine.eckhardt@nilu.no)

**Abstract.** Existing Lagrangian particle dispersion models are capable of establishing source-receptor relationships by running either forward or backward in time. For many applications, backward simulations can be computationally more efficient by several orders of magnitude. However, to date, the backward modelling capabilities have been limited to atmospheric concentrations or mixing ratios. In this paper, we extend the backward modelling technique to substances deposited at the Earth's surface by wet scavenging and dry deposition. This facilitates efficient calculation of emission sensitivities for deposition quantities, which opens new application fields such as the comprehensive analysis of measured deposition quantities, or of deposition recorded in snow samples or ice cores. This could also include inverse modelling of emission sources based on such measurements. We have tested the new scheme as implemented in the Lagrangian particle dispersion model FLEXPART v10.2 by comparing results from forward and backward calculations. We also present an example application for black carbon concentrations recorded in Arctic snow.

## 1 Introduction

Lagrangian particle dispersion models (LPDMs) are popular tools for simulating the dispersion of trace gases, aerosols or radionuclides in the atmosphere (e.g., Stohl et al., 1998; Lin et al., 2003; Witham et al., 2007; Stein et al., 2015). LPDMs typically consider only linear processes, i.e., processes that do not depend on the concentration of the simulated tracer such as non-linear chemical reactions. (Thomson, 1987; Flesch et al., 1995). There are three major reasons for backward simulations: Firstly, in the case where the number of source elements ($I$) is larger than the number of receptors ($L$), running the model backward from the receptors is computationally more efficient than running it forward from the sources. For instance, if one is only interested in the model result for one ($L$=1) receptor point (e.g., a measurement station), a backward simulation of a single tracer is sufficient, regardless of the number of sources $I$. By contrast, in forward mode, $I$ simulations would be needed to obtain the same information at the receptor, which – depending on $I$ - can be orders of magnitude computationally less efficient (Seibert and Frank, 2004). Secondly, particle transport in a LPDM does not require a computational grid and thus dispersion calculations can be initialized from a point location. This property allows backward simulations to be made exactly from the point where a measurement (e.g., an in-situ observation of a trace gas or aerosol) is made, leading to higher accuracy than corresponding forward simulations. For the latter, a kernel or averaging volume is needed that is large enough to





contain a sufficiently large number of particles for robust evaluation of the receptor concentrations. Thirdly, the output of a backward simulation is equivalent to an emission sensitivity and can be used conveniently and without further data processing to visualize which regions potentially influenced the receptor point, or to quantitatively map the source contributions in case the emissions are known (Stohl et al., 2003).

For a linear model, the source-receptor (s-r) relationship is a matrix $M$ whose elements $m_{il}$ can be written as

$$m_{il} = \frac{y_l}{x_i} \tag{1}$$

where $y_l$ ($l = 1,... L$) are the receptor quantities (e.g., concentrations, mixing ratios, or deposition values) and $x_i$ ($i = 1,…$ $I$) are the sources (e.g., emission fluxes). $M$ is provided by the dispersion model and describes the linear relationship between the sources and the receptors. The model must be set to correctly produce the physical units of $M$

corresponding to the units with which the source and receptor quantities are given (e.g., source strengths as kg, receptor values as concentrations in kg m$^{-3}$, or deposition values in kg m$^{-2}$). This was discussed in detail by Seibert and Frank (2004).

Calculation of $M$ with a LPDM has two distinct advantages: 1) The model needs to be run only once. When $M$ is known, the influence of changing the sources $x_i$ (e.g., using different emission scenarios) on the receptor values $y_l$ can

be calculated easily without re-running the dispersion model. 2) It is not relevant whether $M$ is obtained from forward or backward simulations and one can conveniently choose the computationally more efficient and/or more accurate option. The first advantage is critically important for inverse modelling (a popular method to "improve" or "optimize" emission data sets based on ambient measurements), as it means that $M$ does not need to be re-calculated in the inversion. The second advantage can make inverse modelling based on LPDM output even more attractive (see, e.g.,

Stohl et al., 2009; Thompson and Stohl, 2014), as the Lagrangian models can accurately and efficiently be run backward from point measurements.

The theory describing s-r relationships in forward and backward mode has been laid out in detail by Seibert and Frank (2004). They also showed simulation examples for the LPDM FLEXPART (Stohl et al., 1998, 2005). However, their practical implementation of the theory into the model for backward runs was completed only for air concentrations

and mixing ratios at the receptor. Seibert and Frank (2004) did not include deposition quantities at the receptor, although they write that "it is easy to extend the methods presented to this case". Yet, such an implementation for running the model backward from deposition quantities does not exist to date, neither for FLEXPART nor, to our knowledge, for any other LPDM. Nevertheless, such a feature is highly desirable for a number of applications. For instance, wet deposition measurements of acidifying compounds are routinely made in air quality networks (Tørseth

et al., 2012). Deposition of dust is measured regularly in ice cores (Bory et al., 2002). Contamination of snow by black carbon (BC) aerosols has recently received a lot of interest because of the potential impact of BC on the snow albedo (Qian et al, 2015). To interpret such measurements and study the sources of the deposited substance, it would be convenient to have a model that is capable of efficient s-r relationship calculations. For instance, many decades of detailed s-r relationships for ice cores could be calculated with relative ease, which would represent a milestone for

the interpretation of ice core data. In this paper, we present an extension of the LPDM FLEXPART that allows such



calculations and test its performance for both dry and wet deposition by performing thorough consistency test between forward and backward runs.We also briefly present a representative application for black carbon concentrations recorded in high latitude snow samples.

5 **2. Implementation**

We implemented the backward option to calculate deposition values in FLEXPART version 10.2. FLEXPART is an open-source (see www.flexpart.eu) LPDM described and validated in detail in Stohl et al. (1998, 2005) and used in many other studies. The reference version of the model can be driven either with meteorological input data from the European Centre for Medium-Range Weather Forecasts (ECMWF) or the National Centers for Environmental Prediction (NCEP). For this study, we used three-hourly ERA-Interim re-analysis data from ECMWF with a resolution of 1˚ latitude x 1˚ longitude and 61 vertical levels. There exist other versions of FLEXPART such as for the Weather Research and Forecasting (WRF) model (Brioude et al., 2013) or the Norwegian Earth System Model (NorESM)/Community Atmosphere Model (CAM) (Cassiani et al., 2016), in which our changes are not yet implemented. Of some importance for this paper is the fact that FLEXPART uses an internal terrain-following coordinate system that is automatically adjusted to the vertical resolution at which meteorological input data in native model coordinates are provided. This coordinate system is set up based on the first meteorological data set read into the model and then fixed in time. As the first meteorological input data set read into the model is different in forward and backward simulations, this can create small differences due to slightly different interpolation (notice, though, that both options are equally accurate). To avoid such numerical inconsistencies, when testing the agreement of backward/forward results, we copied the internal coordinate system of the forward simulations to the backward runs.

FLEXPART advects particles according to the large-scale winds, superimposed with random velocities representing turbulence as described in Stohl and Thomson (1999) and and Cassiani et al. (2015) and unresolved mesoscale motions as describe in Stohl et al. (2005). FLEXPART also has a deep convection scheme (Forster et al., 2007) which was not used for the forward/backward consistency test simulations, as it is known to introduce small but unavoidable differences between the two simulation settings. (Seibert and Frank, 2004). FLEXPART can take into account dry and wet deposition of gases or aerosols, gravitational settling of particles and radioactive decay (Stohl et al., 2005). Our simulations use the recently implemented and tested wet deposition scheme of Grythe et al. (2016).

FLEXPART produces gridded model output (which can be interpreted as concentration or deposition values in forward runs and emission sensitivities in backward runs) using a uniform kernel (Stohl et al., 2005). The kernel effectively transforms information from the individual particles onto a regular grid, thereby reducing the stochastic errors caused by the limited number of particles residing in a given volume compared to simple volume counting. However, to allow direct comparisons of forward and backward simulations, particles need to be released and sampled in exactly the same volumes. Since the kernel cannot be used for the particle release, we also do not use it here for producing the model output. Instead, we release particles in latitude-longitude grid boxes and determine the concentrations in





identical boxes by summing the mass of all particles in a box and dividing by the box volume. This ensures direct comparability between backward and forward simulations. Notice that for other applications than forward/backward comparisons, the use of the kernel is preferable.

For determining deposition amounts in forward simulations (deposition is always given in accumulation as kg m$^{-2}$),

we use FLEXPART in its standard configuration, which calculates two-dimensional wet and dry deposition fields in addition to the three-dimensional concentration fields. The determination of the s-r relationship for deposition quantities in backward mode is done separately for wet and dry deposition. For dry deposition, particles are released in the receptor grid cell within a shallow layer adjacent to the ground. At the time of the release, each particle's "mass" is multiplied with the local dry deposition velocity. For wet deposition, particles are released over the entire

atmospheric column, as scavenging can occur at any height of the atmosphere, depending on the location of clouds and precipitation. At the time of the release at the receptor point, a deposition velocity (m s$^{-1}$) is assigned to every particle. For the dry deposition, the velocity is directly calculated by FLEXPART's dry deposition scheme. For the wet deposition, the deposition velocity is obtained by multiplying the local scavenging coefficient (unit of s$^{-1}$, see FLEXAPART user manual) with the release altitude (unit of m). After the particle release, both for dry and wet

deposition, particles are tracked as in a standard FLEXPART backward simulation configured to obtain the concentration at the receptor point expressed in mass per volume. We recall that in a such configured standard backward run the s-r relationship has units of s$^{-1}$ or s kg m$^{-3}$ depending on the units chosen for the source emission (see Table 1). Therefore, in the deposition runs these units are multiplied with the deposition velocity (see Table 1). The s-r relationship obviously also includes the treatment of all loss processes, including wet and dry deposition,

occurring en route. To obtain the s-r relationship for total (dry plus wet) deposition, the s-r relationships for wet and dry deposition must be calculated individually in separate FLEXPART runs and added in post-processing.

Table 1 is an extension of Table 2 in Stohl et al. (2005) and reports the units used for forward and backward calculations, where the entries for the deposition calculations are new. The user settings required to produce the simulations are reported as well. In addition to the user input and output units, we also report the unit of the s-r

relationship. This is equivalent to the values given in Table 1 of Seibert and Frank (2004), however, with two important differences: 1) Emissions are assumed to be given in kg or as a mixing ratio in forward mode and in kg m$^{-3}$ s$^{-1}$ in backward mode, as these are the most commonly used options in FLEXPART. Seibert and Frank (2004) reported values for emissions given both as rates per time and as totals, for both forward and backward runs. 2) In backward mode, input quantities are considered unitless, as the s-r relationship is scaled with the inverse of the input quantity.

This is done in order to avoid a dependence of the model output on the "release mass" value used as input. Thus, the model output of a backward simulation can be multiplied directly with the emissions (in kg m$^{-3}$ s$^{-1}$) in order to obtain the desired concentration, mixing ratio or deposition quantity at the receptor. All this is identical to the previous treatment in FLEXPART (Stohl et al., 2005), except for the addition of the deposition options.





## 3 Evaluation

### 3.1 Grid-scale performance

To test the implemented algorithm we modelled 24 hours of dispersion, dry and wet deposition after an emission of black carbon (BC) in one grid cell (marked with a black rectangle in Fig. 1) over 1 hour. BC was selected as a tracer here because it is subject to both wet and dry deposition. In forward mode, representing an emission of an arbitrary amount of 100 kg of BC, 1 million particles were released on 18 March 2012 between 15:00 and 16:00 UTC in a 1°x 1° x 100 m box centered at 4.5°E, 69.5°N. BC concentrations and deposition values were evaluated in three receptor boxes, one identical to the emission box, one at one grid cell distance (5.5°E, 70.5°N), and one at two grid cells distance (6.5°E, 67.5°N) from the release box (black circles in Fig. 1). The concentration averaging time was 1 hour. For the backward simulations, particles were released in the receptor boxes (for concentrations), in 30-m high boxes (for dry deposition) and in the atmospheric column (for wet deposition) over the receptor cells, and during 24 intervals of 1-hour corresponding to the sampling times of the forward simulation. Two hundred thousand particles were released for the dry deposition/concentration backward run and 2 million particles for the wet deposition backward run. To ensure a correct simulation of turbulence and dry deposition on such small scales, we limited the model's numerical time step to 10% of the Lagrangian time scale (FLEXPART setting CTL=0.1). For determining output quantities (concentrations, depositions, emission sensitivities), grid-cell sampling of the particles was performed every 90 s. Notice that for this point-source set-up and wet deposition the forward mode is computationally more efficient than the backward mode since the receptor extend over the atmospheric column.

In the forward simulation, the BC tracer spreads eastwards and quickly reaches the evaluation grid cells (Fig. 1). The concentration and dry deposition patterns (Fig. 1, top and middle row) are very similar, as deposition velocity only modulates the surface concentration pattern. The wet deposition pattern (Fig. 2, bottom row) is different, as it depends on tracer concentrations in the entire atmospheric column and only occurs when there is precipitation.

Figure 2 shows a comparison of the results from the forward and backward simulations obtained for the three receptors (distinguished by different colours), both as time series (left panels) and scatter plots (right panels). In the time series plots, the lines and circles show the results from the forward simulation, while the black stars show the results from the backward simulation. In the first receptor box, which is identical to the emission box (black line and symbols), concentration and deposition values peak right at the time of the emission, with a secondary peak at around hour 15, while wet deposition peaks at hour 2 as there was no precipitation during the first hour. In the second receptor box, concentration and dry deposition peaks in the middle of the 24-hour period considered, while in the third receptor box, all quantities peak at the end. For all three quantities (concentrations, dry and wet deposition), the temporal behavior is very similar in the forward and backward simulations and values are similar, though not identical.

The scatter plots (right panels) show the relative differences between the forward and backward model simulations as a function of the simulated forward quantity. To facilitate plotting of the results for all three grid cells on the same scale, values shown on both axes are normalized with the maximum value obtained in the forward simulation. Ideally, all points should lie on the yellow horizontal line, which is not the case in our simulations. The black dotted lines indicate forward-backward differences of 10 and 20%, respectively. Most of our values lie within the 20% lines but



there are a few outliers with larger errors. Table 2 summarizes these results. The relative errors are somewhat larger than those shown in Seibert and Frank (2004) for concentration values, but they considered a more simple case.

As already described in Seibert and Frank (2004), it cannot be expected that the simulations match perfectly. The distribution of the particles in the grid cell of the release (both forward and backward) is not perfectly homogeneous and initial position differences can be amplified by atmospheric transport. Some of these errors are of stochastic nature and decrease with the number of particles used. However, our particle numbers are very large and tests have shown that further increasing these numbers did not improve the agreement. Furthermore, the same simulations were performed with a passive tracer and the results for the concentrations are as good as those for the BC species. We have also repeated the simulations with the turbulence parameterizations switched off and obtained similarly different results. This demonstrates that the differences arise mainly from the interpolation of the grid-scale winds, with small initial position errors being enhanced during transport. Importantly, however, the relative errors for wet and dry deposition are not larger than those for the concentration values. This shows that our implementation of backward modeling capabilities for wet and dry deposition does not introduce additional errors.

### 3.2. Long-range transport performance

FLEXPART is often used in backward mode to find the source regions for specific concentration measurements of substances with globally distributed sources (e.g. for BC, $CH_4$; see Stohl et. al 2013; Thompson et al., 2017). With the described algorithm, this is now also possible for deposition measurements. We used BC emissions (assumed annually constant) from the ECLIPSE (Evaluating the Climate and Air Quality Impacts of Short-Lived Pollutants) V5 global inventory of anthropogenic emissions (Stohl et al., 2015; Klimont et al., 2016) and GFED (Global Fire Emission Database) biomass burning emissions (Giglio et al., 2013). Using the emission inventory at the full 0.5° resolution with a limited total number of particles can, in regions with low emissions, lead to large relative uncertainties in simulation results, due to errors in the discretization of the emissions. Therefore, the emission fluxes were averaged over 3° x 3° areas and only 96 emission grid cells over Europe were used in the simulation, yielding a simplified but still realistic emission scenario. Two million particles were released in every emission grid cell in the forward simulation giving a total of 192 million simulated particles.

The simulations were performed for two months, where the first month was used for spin up. The results of forward and backward calculations were compared for the second month (March 2012) and for two receptor points. Computational time steps were allowed to exceed the Lagrangian time scale (FLEXPART option CTL=-5), which is often done for large-scale FLEXPART simulations, and particle sampling was performed every 900 s. The backward simulations were performed every 3 hours, releasing 50000 particles for the concentration and dry deposition calculation and 200000 for the wet deposition calculation, during each 3-hour interval giving a total of 12 million simulated particles.

Figure 3 shows the emission flux of the inventory used (Fig. 3a), the average concentration in the lowest model layer (0-100 m above ground level) (Fig. 3b), and the accumulated wet (Fig. 3c) and dry deposition (Fig. 3d) for March



2012. Based on these results, we selected two locations where we compare the results for forward and backward simulations (Table 2). The two points represent very different concentration and deposition levels, due to their different distances from strong BC source regions. While point A is located relatively close to strong emission sources (average concentration of 270 ng m$^{-3}$), point B on Spitsbergen in the Arctic is far away from sources (average

concentration of 7 ng m$^{-3}$) .

In general, the forward and backward simulations show very good agreement for both receptor points. For example, the distinct daily cycles in concentration and dry deposition at point A are simulated similarly, and the mean concentration and deposition values are almost identical in the forward and backward simulations at both points.

However, during some episodes there can be notable differences, for example at the end of the simulation period at point A.

The summary in Table 3 shows that there is no systematic bias between the forward and the backward simulations. The mean differences between forward and backward simulations are between -8.4 and 13%, for concentrations and

depositions. There is also no evidence that biases are systematically larger for deposition values than for concentration values, indicating that no systematic errors have been added by implementing the backward modelling capabilities for deposition values. Backward and forward modelling results agree with each other within 20% in 37-86% of all cases. Again, there is no evidence that relative errors are larger for deposition quantities than for concentration values.

**4. Source regions for modelled BC deposition**

A nice feature of the backward simulations is that the model output is an emission sensitivity (ES) that shows the regions for potential emission uptake (see Table 1). Most emissions are released at or near the surface and we therefore calculate the sensitivity for the lowest 100 m above ground. When folding the ES with the emission fluxes from an emission inventory (see Fig. 3a), we obtain the emission contribution for each grid cell. The total value at the receptor

location is then obtained by area integration of the emission contributions.

To give an example of the emission sensitivity and emission contribution plots, we calculate their average values during the period of 8$^{th}$ – 10$^{th}$ of March 2012 for receptor point B, when there was a peak in both dry and wet deposition (Fig. 6a). The emission sensitivity is similar close to point B for both wet and dry deposition but values for wet deposition are higher and also include regions not present in the dry deposition sensitivity. For example, the high

values of the emission sensitivity over the North Atlantic must be due to uplift and arrival of air masses at greater altitude at point B. Consequently, emission contributions, while generally similar for dry and wet deposition, include larger areas for the wet deposition (e.g., over Great Britain) than for the dry deposition. Simulated total wet deposition is also much larger for wet (35 kt in March) than for dry (5 kt in March) deposition, which is typical for BC.





### 5. Comparison with BC concentration measurements in snow

As a practical example, we used our method to compare modelled concentrations of BC in snow with measurements taken all over the Arctic (Alaska, Canada, Greenland, Svalbard, Norway, Russia, and the Arctic Ocean) from 2005 to 2009 (adopted from Doherty et al. (2010)). Of course, these measurements are only indirect measurements of BC

deposition, as the BC snow concentration also depends on the amount of snow fall and can also be influenced by post-depositional processes in the snow pack. To minimize the latter effects, we used only those samples that were collected in spring, before the snow had started to melt, in order to avoid percolation effects of the meltwater through the snowpack. Although the measurements are indirect, we consider this as a typical application example for the new FLEXPART feature. We performed a backward simulation for every measurement sample, where the ending time of

the particle release was set as the date and time when a snow sample was collected. The beginning time of the particle release was set as the time when precipitation from ECMWF had accumulated, backwards in time from the sampling time, the same amount of water as the water equivalent of the snow sample up to the specified sampling depth. This procedure assumes that the ECMWF precipitation is a good proxy for real snow accumulation, an assumption that introduces additional uncertainties that may be of a similar magnitude as those in the simulated BC deposition. We

calculated the sum of dry and wet deposition for each sample, as the measurements do not allow distinguishing the two contributions.

To assess the obtained results, we calculated the relative difference between modelled and observed snow BC by means of the mean fractional bias (MFB), which is defined as:

$$MFB = \frac{1}{N} \sum_{i=1}^{N} \frac{C_m - C_o}{(C_m + C_o)/2} \times 100\% \qquad (2)$$

where $C_m$ and $C_o$ are the modelled and observed BC concentrations in snow [ng m$^{-2}$] and $N$ is the total number of observations. This statistical measure is a useful model performance indicator because it gives the same weight to under- and overestimations (values range between -200% and 200%). The results were accurate in most of the Arctic regions (MFB=-51%) except for the Canadian Arctic, where snow concentrations predicted by the model in 2007 were

one order of magnitude higher than the measurements. Further analysis of this overestimation showed that the air was coming from continental regions of Canada, where boreal forest fires burned at the time when sampling took place. The model seems to overestimate their influence on the snow BC concentrations, possibly due to too coarse temporal resolution of the GFED fire emissions. Samples from the same regions in other years showed good agreement with modelled values. If the large mismatch in the Canadian Arctic in 2007 is removed, the underestimation is, however,

more significant (-36%). This could suggest an underestimation of BC emissions in the major source regions of Arctic BC. The RMSE (root mean square error) was estimated to be 17 mg m$^{-2}$, which is acceptable considering that the measured snow concentrations in the dataset ranged from 18 to 244 mg m$^{-2}$. The highest concentrations of snow BC were observed over Russia, where the model showed a good agreement. Excluding the 2007 values from this comparison, the RMSE was about 9 mg m$^{-2}$. For instance, the highest values were obtained in Western Siberia, close

to the gas flaring regions of the Nenets/Komi region, as well as in South-eastern and North-eastern Russia, where air masses were arriving from strong emission sources in South-eastern Asia.





The highest emission sensitivities for the snow samples are in the polar region above 70 °N (Fig 8a). Combining this with the emissions gives the modelled source contributions per grid cell, which results in an average modelled contribution of 25.1 ng g$^{-1}$ for all samples. The flaring area south of the Yuzhny island sticks out as an important contributor. Besides this, the BC measured in the snow samples originates from Canadian biomass burning emission, European anthropogenic emissions and Asian anthropogenic emissions.

It is beyond the scope of this paper to analyse the snow BC data in more detail. However, it is clear that plots like those shown in Fig. 8, either for individual snow samples or also as averages for groups of samples (e.g., for groups of high versus low measured concentrations), can strongly support the interpretation of snow BC concentration measurements. With the new modelling capabilities of FLEXPART such plots can be produced easily.

## 6. Conclusions

In this paper, we have described a substantial extension of the backward modelling capabilities of the Lagrangian particle dispersion model FLEXPART. The model is now capable of calculation of source-receptor relationships in backward mode also for wet and dry deposition. To date, such calculations were only possible for air concentrations. To our knowledge, the new model capabilities are unique to FLEXPART. The calculations are extremely efficient, with computation times depending almost exclusively on the number of receptor points. Therefore, the new method is suitable for establishing detailed source-receptor relationships also for long-term records of deposition, such as ice core or sediment records. The method can be applied to all substances that are removed from the atmosphere by dry or wet deposition and are not subject to strong non-linear chemistry in the atmosphere. Prominent examples are dust or black carbon records and, with some limitations with respect to non-linearity, deposition of eutrophying substances.

We have tested the method by comparing wet and dry deposition values obtained from forward and backward simulations. This was done in a small-scale synthetic case study and also for a realistic long-range transport application of the model at the example of BC. In the latter study, we considered two receptor points, one in a remote area and one close to strong emission sources. In both studies, we find good agreement between the forward and backward calculations. In the short-range transport case study, we find that 60% of the values agree within 10% and 70% of all values agree within 20%. In the long-range transport case study, we find that 41% of the values agree within 10% and 59% of all values agree within 20%. The differences are due to stochastic noise (e.g., due to the initialization of particle positions) and, especially, the interpolation of grid-scale winds, which can be amplified during transport. To limit stochastic errors in the discretization of emissions and obtain statistically robust concentrations, a very large number of particles is needed in the forward simulation, again demonstrating the relative efficiency of backward simulations if one is interested in the model results only at a small number of locations (e.g., measurement sites).

We also demonstrated how plots of emission sensitivity and emission source contributions can help with the analysis of the sources contributing to a simulated deposition value. Finally, in a first application of the new method, we also



presented a comparison of simulated and measured BC concentrations in snow, using a measurement data set covering many locations in the Arctic.

## 7. Code and data availability

The FLEXPART model can be downloaded from http://www.flexpart.eu. Any FLEXPART code is free software
5   distributed under the GNU General Public License and it is maintained using the Git system.

The version described here can be downloaded by using "git clone *https://www.flexpart.eu/gitmob/flexpart"*, accessing the tagged version 10.2: "git checkout v10.2beta". The model results discussed here are also available upon request.

10   **Acknowledgement**

The authors acknowledge S. J. Doherty for providing the database of snow BC observations described in Doherty et al. (2010). We thank Zbigniew Klimont and Chris Heyes at the International Institute for Applied System Analysis – IIASA for providing BC emissions from their GAINS model, ECMWF is acknowledged for meteorological data and Louis Giglio and Guido van der Werf for the GFED data. Computational and storage resources for FLEXPART
15   simulations were provided by NOTUR (NN9419K) and NorStore (NS9419K). Funding was received as part of eSTICC-eScience Tools for Investigating Climate Change in northern high latitudes, which is supported by Nordforsk Nordic Centre of Excellence grant 57001. Andreas Stohl and Massimo Cassiani were supported by the European Research Council (ERC) under the European Union's Horizon 2020 research and innovation programme under grant agreement No 670462 (COMTESSA).

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



| IND_SOURCE | IND_RECEPTOR | Source | Receptor | mode | s-r unit | input unit | output unit |
|---|---|---|---|---|---|---|---|
| 1 | 1 | mass | mass | fwd | $m^{-3}$ | kg | $ng\ m^{-3}$ |
| | | | | bwd | s | 1 | s |
| 1 | 2 | mass | mix | fwd | $kg^{-1}$ | kg | pptm |
| | | | | bwd | $s\ m^3\ kg^{-1}$ | 1 | $s\ m^3\ kg^{-1}$ |
| 2 | 1 | mix | mass | fwd | $ng\ m^{-3}$ | 1 | $ng\ m^{-3}$ |
| | | | | bwd | $s\ kg\ m^{-3}$ | 1 | $s\ kg\ m^{-3}$ |
| 2 | 2 | mix | mix | fwd | 1 | 1 | pptm |
| | | | | bwd | s | 1 | s |
| 1 | 1 or 2 (deposition output) | mass | mass depo | fwd | $m^{-2}$ | kg | $ng\ m^{-2}$ |
| | 3 (wet) or 4 (dry) | | | bwd | m | 1 | m |
| 2 | 1 or 2 (deposition output) | mix | mass depo | fwd | $ng\ m^{-2}$ | 1 | $ng\ m^{-2}$ |
| | 3 (wet) or 4 (dry) | | | bwd | $kg\ m^{-2}$ | 1 | $kg\ m^{-2}$ |

*Table 1: Extension of Table 2 from Stohl et al. (2005) to deposition quantities. The table reports the input and output units used for FLEXPART forward and backward simulations. IND_SOURCE and IND_RECEPTOR are the values chosen in FLEXPART's "COMMAND" file to produce the corresponding simulation. Notice that in backward simulations the release takes place at the receptor and the sampling at the source. In forward mode, the deposition output is always provided in mass units and no specific user setting is needed, as the deposition output is made in addition to the concentration (or mixing ratio) output. See further explanations in the main text. In the columns s-r unit and input unit, the number 1 means dimensionless.*



|  | within 10% | within 20% |
|---|---|---|
| Conc. | 48.6 | 68.1 |
| Dry depo. | 44.4 | 65.3 |
| Wet depo. | 76.4 | 77.8 |

**Table 2: Relative frequency in percent of concentration and deposition values with relative errors smaller than 10% and 20%.**



| Receptor | Type | Mean fwd | Mean bias % | within 10% | within 20% |
|---|---|---|---|---|---|
| A | Conc. | 262.33 | 8.43 | 36 | 73 |
| A | Dry depo. | 1441.53 | 12.89 | 23 | 50 |
| A | Wet depo. | 6675.23 | -11.53 | 80 | 86 |
| B | Conc. | 7.46 | -9.08 | 23 | 41 |
| B | Dry depo. | 47.91 | 11.40 | 19 | 39 |
| B | Wet depo. | 816.52 | -3.85 | 43 | 59 |

**Table 3: Statistical comparison of forward and backward simulations for concentrations, dry deposition and wet deposition values, for receptor points A and B. Reported are the mean values for March 2012 for the forward simulation, the mean relative bias between backward and forward simulations (%), and the percentage of all values (out of 248 in total) of the backward simulation with less than 10% (20%) deviation from the forward simulation.**





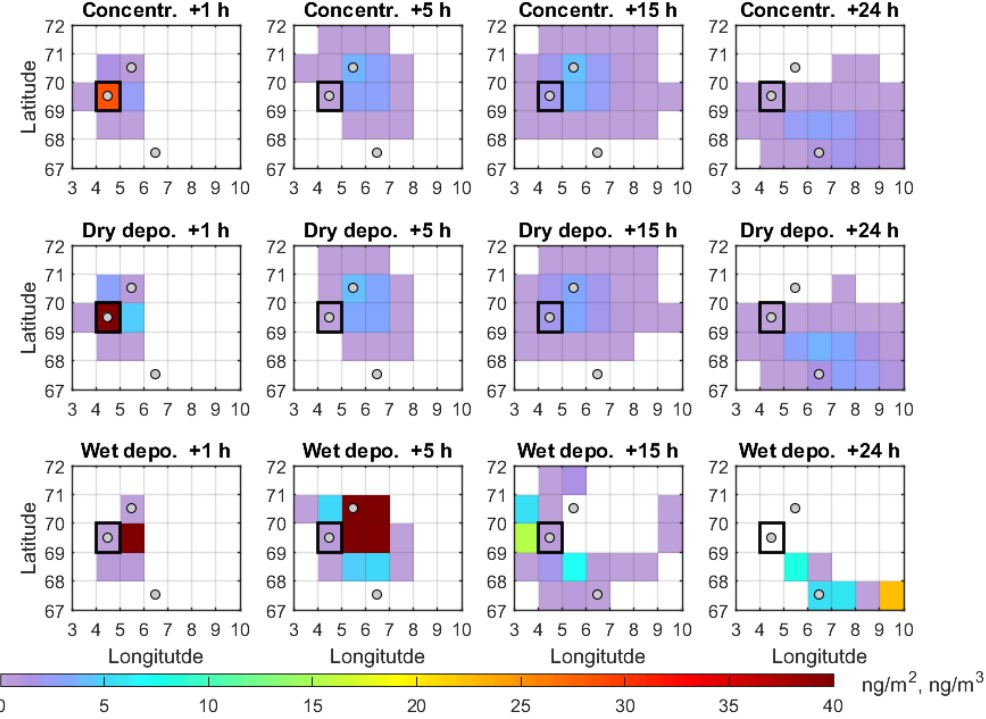

**Figure 1: Concentration of BC (ng m$^{-3}$) in the lowest 100 m of the atmosphere (top panels), dry deposition (middle panels) and wet deposition (bottom panels) (both in ng m$^{-2}$) resulting from a 10 kg BC emission in the grid cell marked with a black rectangle. Grid cells are shown with black horizontal and vertical lines. The panels show (from left to right) the results 0-1, 4-5, 14-15 and 23-24 hours after the start of the emission. The shaded grey circles mark grid cells for which the forward model output (as shown here) was compared with backward model simulations, i.e., from where particles were released in backward mode.**




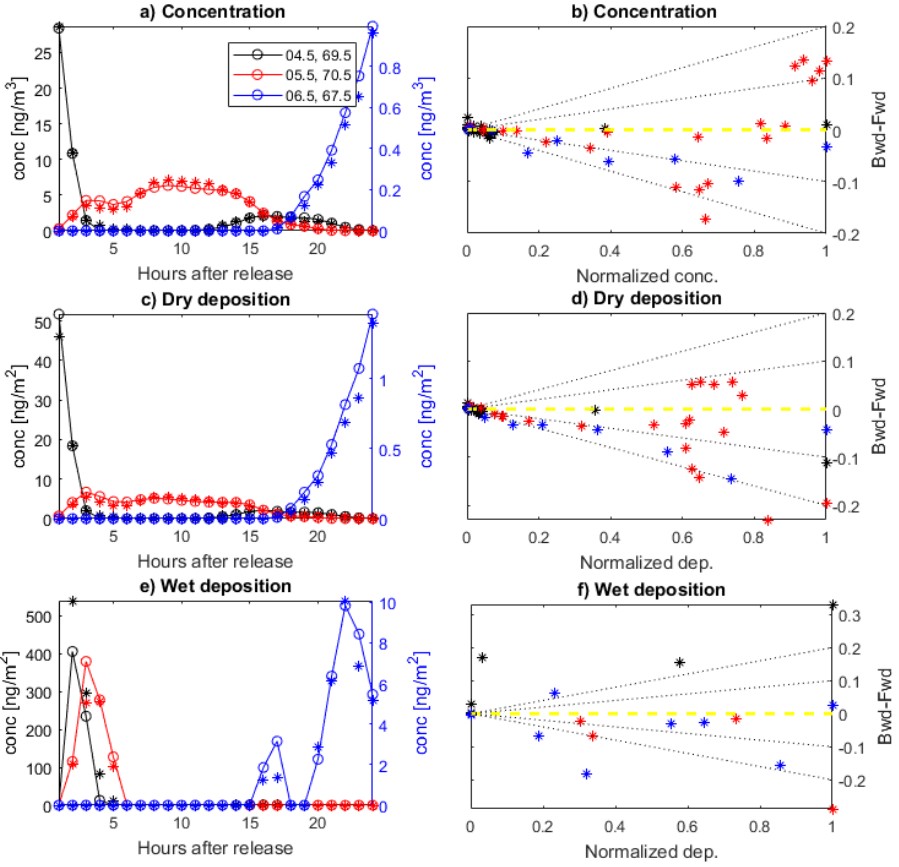

**Figure 2: Comparison of modelled concentration and deposition values from forward and backward simulations for the three receptor grid boxes (1) black: 4.5° E, 69.5° N; (2) red: 5.5° E; 70.5° N; (3) blue: 6.5° E, 67.5° N). The left panels show the data as time series for the 24 hours after the emission. The stars depict the values obtained by the backward simulation and the circles and lines the corresponding values from the forward simulation. Notice that due to the lower simulated values at receptor grid box number three , the scales are different for this box and are reported on the right axis in blue colour . The panels on the right hand side show the normalized difference between backward and forward (Bwd-Fwd) simulations as a function of the simulated value obtained with the forward simulation and normalized by the maximum simulated value in the forward simulation. Black dotted lines correspond to 10 and 20% relative error, respectively.**





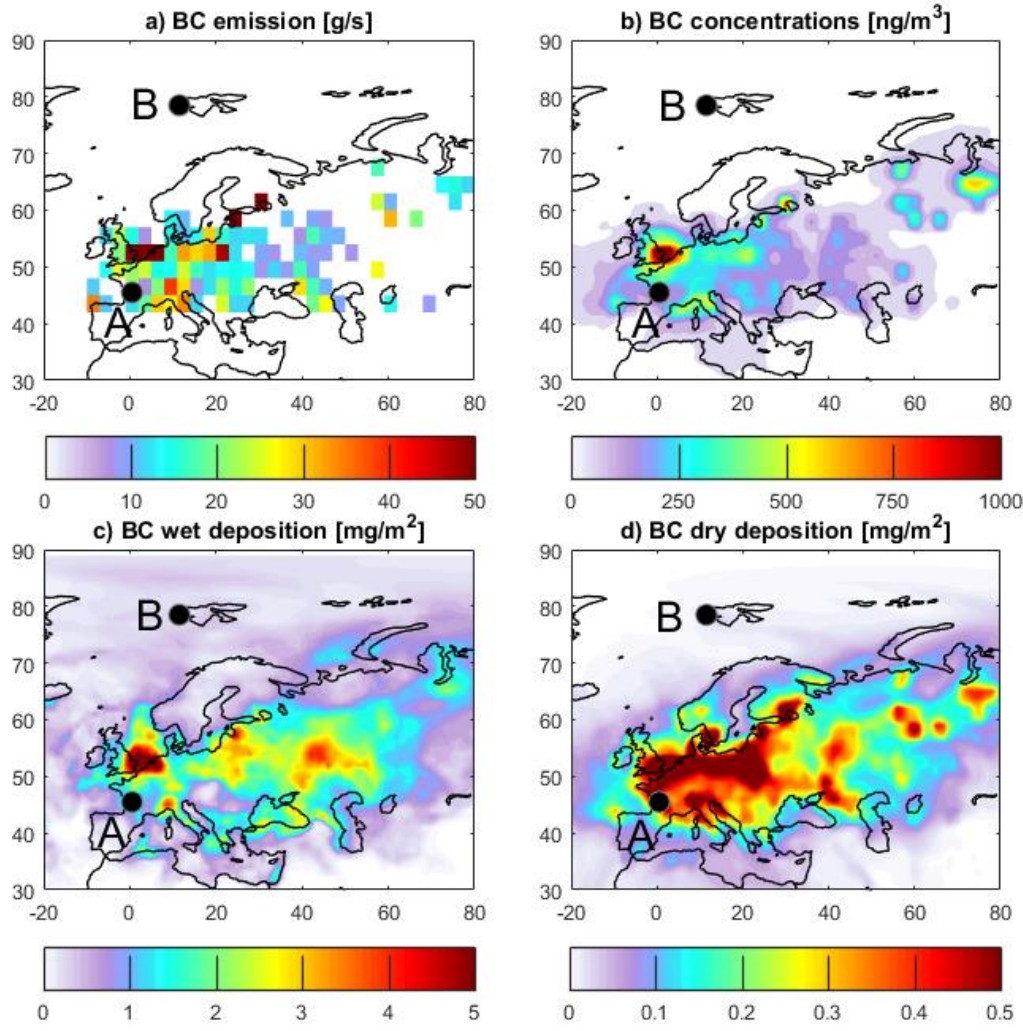

**Figure 3: Average BC emission fluxes (a), average BC concentrations in the lowest model layer (0-100 m) (b), accumulated BC wet deposition (c) and accumulated BC dry deposition (d) for the period of 1 March to 1 April 2012. The black dots show the locations A and B for which a detailed comparison of forward and backward calculations is performed.**





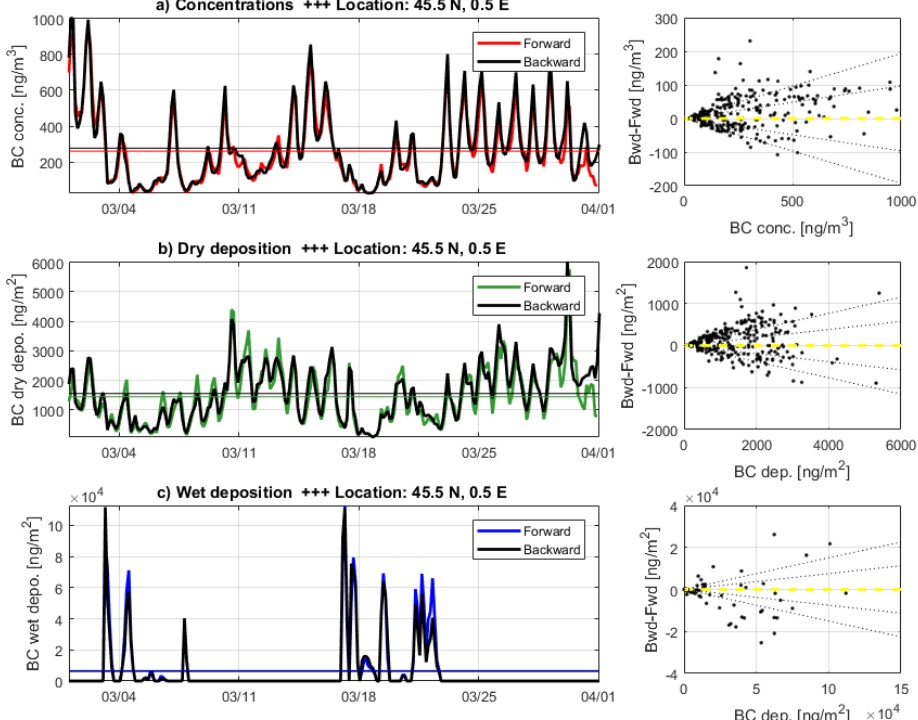

**Figure 4: Average 3-hourly BC concentrations in the lowest 100 m (panel a), BC dry deposition (panel b) and BC wet deposition (panel c) from forward (red, green, blue) and backward (black) simulations for the receptor point A (shown in Fig. 3) for March 2012. The two horizontal lines show the mean value of the forward and the backward run for the whole period. Panels on the right hand side show the difference between the backward and the forward simulation versus values from BC concentration/deposition of the average of backward and forward simulation.**





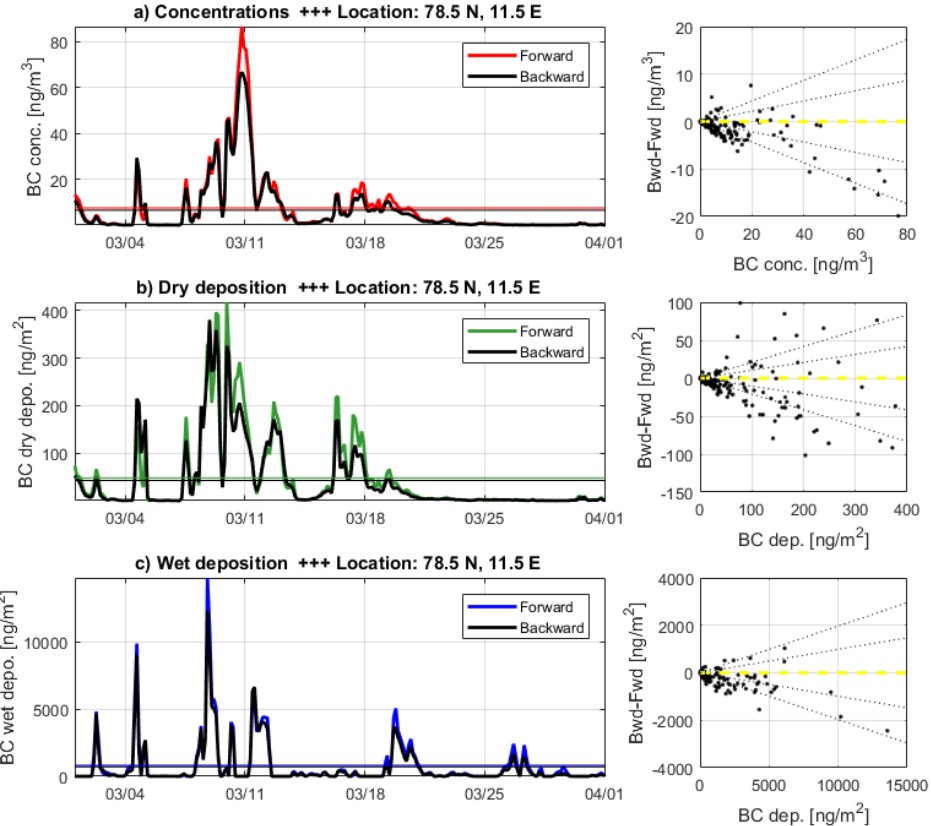

**Figure 5: Same as Figure 4, but for the receptor point B.**



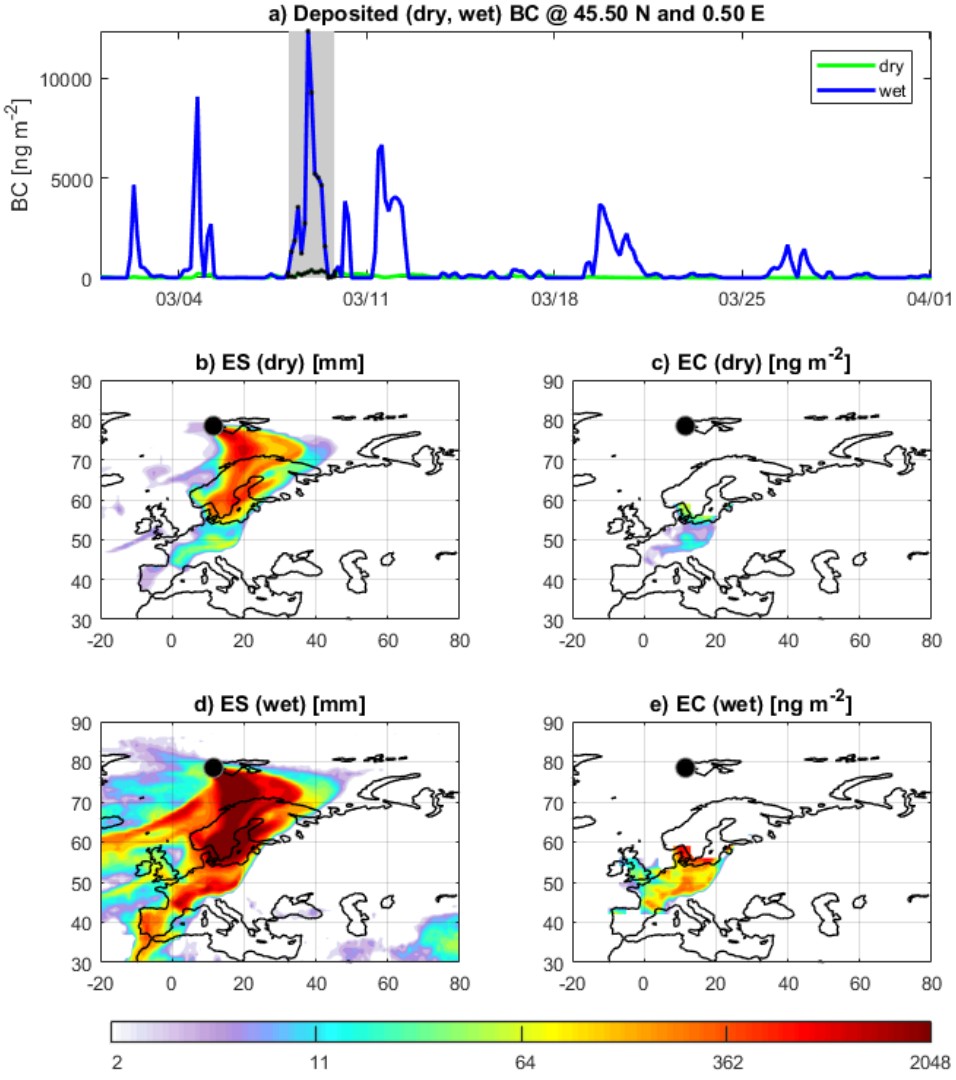

**Figure 6: Emission sensitivities and emission contributions for point B averaged over the period 8-10 March 2012. Panel a shows time series of backward modelled dry (green line) and wet deposition (blue line) for point B. In panels b and c, the average emission sensitivity (ES) [mm] and emission contribution (EC, i.e., emission sensitivity multiplied with emission fluxes from Fig. 3a) [ng m⁻² per grid cell] are shown for the dry deposition during the period marked with black dots and grey background shading in panel a. Panels d and e show the same as panels b and c, but for wet deposition.**





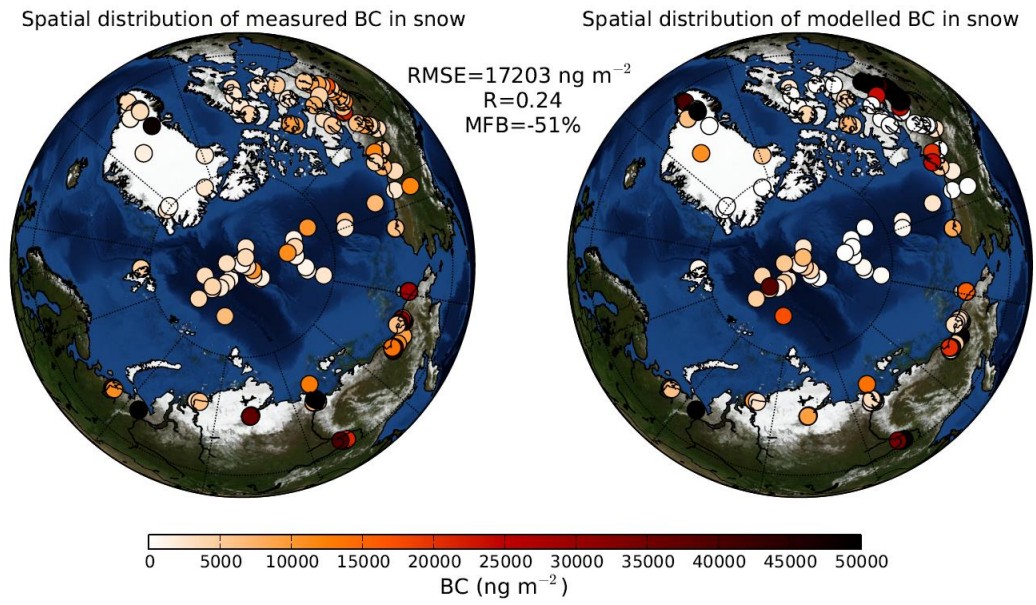

**Figure 7: Measured and modelled BC concentration in snow.**





**Figure 8: Average footprint emission sensitivitiy and source contribution for all samples shown in Fig. 7.**