# Peer review of "Source-receptor matrix calculation for deposited mass with the Lagrangian particle dispersion model FLEXPART v10.2 in backward mode"

_Geoscientific Model Development, 2017_

## Short Comment (SC1) · 14 Jul 2017

Dear authors,

In my role as Executive editor of GMD, I would like to bring to your attention our Editorial version 1.1:

http://www.geosci-model-dev.net/8/3487/2015/gmd-8-3487-2015.html

This highlights some requirements of papers published in GMD, which is also available on the GMD website in the 'Manuscript Types' section:

http://www.geoscientific-model-development.net/submission/manuscript_types.html

In particular, please note that for your paper, the following requirements have not been met in the Discussions paper:

- "All papers must include a section, at the end of the paper, entitled 'Code availability'. Here, either instructions for obtaining the code, or the reasons why the code is not available should be clearly stated. It is preferred for the code to be uploaded as a supplement or to be made available at a data repository with an associated DOI (digital object identifier) for the exact model version described in the paper. Alternatively, for established models, there may be an existing means of accessing the code through a particular system. In this case, there must exist a means of permanently accessing the precise model version described in the paper. In some cases, authors may prefer to put models on their own website, or to act as a point of contact for obtaining the code. Given the impermanence of websites and email addresses, this is not encouraged, and authors should consider improving the availability with a more permanent arrangement. After the paper is accepted the model archive should be updated to include a link to the GMD paper."

I'd like to thank you, as you have put in the effort to make the exact version of the code accessible. Unfortunately GIT systems are not a permanent archive. Therefore I'd like to ask you to make the exact version (FLEXPART v10.2beta) available via a permanent archive providing a DOI (e.g. Zenodo).

Yours,

Astrid Kerkweg

---

## Short Comment (SC2) · 20 Jul 2017

Dear Editor,

Thank you for your comment, also the handling editor David Ham mentioned this as something to be done after the review process. FLEXPART is a community software, it has been openly available for over 20 years. The FLEXPART community cares to make the model easily accessible. For now, I will upload a tar ball with the exact version to the communities website (flexpart.eu), we can also include it as a supplementary material to the manuscript. We will be discussing the DOI option within the community.

Kind regards

Sabine Eckhardt

---

## Referee Comment (RC1) · Anonymous Referee #1 · 27 Jul 2017

In this manuscript, the authors describe an addition to the FLEXPART code which allows for backward trajectory calculations of wet and dry deposited material. They test their scheme by comparing forward and backward trajectory runs, and show a real-world application for black carbon deposition in the Arctic.

This is a potentially useful contribution to GMD, that details a bit of very nice development to the FLEXPART code. So in principle I think that this manuscript could be suited for the GMD special issue.

However, the manuscript would benefit greatly from some careful proofreading and revision of the text and presentation. At times it is somewhat sloppy, making it hard to

follow the authors' thoughts

In particular, I have the following major comments:

1) The order of paragraphs and ideas in the introduction is somewhat haphazard, making for a quite confusing read. I suggest the authors careful lay their ideas in a comprehensive ordering

2) The authors might want to consider adding a schematic or similar to the discussion of their implementation in section 2; that might make it much easier to follow exactly what the authors have done

3) The authors on multiple occasions state that results are 'similar' (e.g. page 5 line 20, line 30). Can this 'similar' be quantified? Is it possible to use some statistical tests to assess whether the two experiments are statistically indistinguishable?

4) Section 5 is at points particularly confusing. I is not clear why a MFB=-51% is considered accurate (page 8, line 18), while an MBF=-36% is considered an underestimation (page 8, line 30). Also this section would likely benefit from a map of these MFBs, as a new Figure perhaps

Furthermore, I also have some minor comments

- In the first paragraph of the introduction, this depends on the interpolation scheme used, I assume?

- Page 2, line 34: calling ones own research a milestone is a bit grand..

- Page 3, line 2: Remove 'briefly' here

- page 3, line 13: How can both interpolation schemes be equally accurate? Do the authors mean 'exactly the same' here? Surely that can't be the case?

- page 3, line 30: This sentence is quite confusing and vague for someone who isn't experienced with using FLEXPART

- page 4, line 8: how shallow is 'shallow' here?

- page 6, line 8-13: Are there any references from the literature here that can be used to back up these claims?

- page 7, line 7: refer to Figure 4 here?

- page 9, line 4: so the sources from Canada are definitely not anthropogenic?

- page 9, line 10: I am surprised there is no source at all from shipping (i.e. over the ocean). Is this a limitation of the technique? Or are there really no Black Carbon sources from shipping?

- Figure 7: What is plotted on the background in this Figure?

- Figure 8left: is the unit here really in nanosecond per cubic meter?

Type-os etc:

- Page 1, line 26 & 32: replace 'Firstly' and 'Secondly' with 'First' and 'Second'

- page 5, line 18: 'extends over the entire atmospheric column'

- page 6, line 6: 'with increasing number of particles'?

- page 7, line 21: Is 'nice' the right word here? Something more technical?

- page 7, line 23: Is 'folding' a type-o?

---

## Referee Comment (RC2) · Anonymous Referee #2 · 7 Aug 2017

This paper presents the addition of wet and dry deposition to FLEXPART applications in backward concentration mode. The manuscript is well written and the information is presented in a clear and concise manner. Nevertheless, I have a major concern: Including wet and dry deposition only for gas species would be perfectly acceptable, however, given that the applications presented in this work involve aerosols the authors missed to incorporate the settling velocity in the backwards application. If the omission is intentional, please include language in your manuscript justifying it.

---

## Short Comment (SC3) · 9 Aug 2017

We thank referee #2 for the positive evaluation of our paper. The referee's only major concern was that gravitational settling is not included in the backward application. However, this is not true. We missed to mention this explicitly (perhaps because settling is not very important for the species considered in the paper), but gravitational settling is in fact included. Thus, we can apply the new method also to particles of larger sizes such as desert dust or volcanic ash.

In FLEXPART, gravitational settling influences two aspects of particle transport and deposition:

1) a settling velocity is calculated at every time step and added to the other velocity components (grid-scale, turbulent) that determine particle transport. In backward mode, settling is reversed and leads to an additional upward drift of a particle. This is already included in backward calculations for (aerosol) concentrations and no additional development was needed.

2) the settling velocity enters the calculation of the dry deposition velocity. This is reversed in the backward calculation as described in the paper. The reversion includes the settling component, although we did not explain that, unfortunately.

In summary, gravitational settling is fully considered, but we realize that we should have explained this explicitly. This will be done in a revised version of our paper.

---

## Author Comment (AC1) · 6 Oct 2017

For BC particles, settling plays a minor (insignificant) role. Therefore, to have a test case with more significant settling, we increased the diameter of the aerosol used in the study to 2 $\mu$m . We performed an evaluation for concentration and dry deposition of these larger particles and added this information to the supplementary material. We added: "For BC used here, the settling only plays a minor role. To test the algorithm also for a substance for which settling is important we made a separate test case focusing on a 2$\mu$m particle. The settling will influence the dry deposition velocity and the concentration. The differences between the forward and the backward simulation

are on the same level as for the BC discussed above. The detailed evaluation can be found in S1."

---

## Author Comment (AC2) · 6 Oct 2017

The authors thank referee 1 for the careful review.
We will address the points (in italic) below:

*1) The order of paragraphs and ideas in the introduction is somewhat haphazard, making for a quite confusing read. I suggest the authors careful lay their ideas in a comprehensive ordering*

We have tried to improve that section. In particular, we now added some text to explain what each paragraph is about to make the structure more clear.

*2) The authors might want to consider adding a schematic or similar to the discussion of their implementation in section 2; that might make it much easier to follow exactly what the authors have done*

This is a good idea. We created a flow chart showing the steps for dry and wet deposition calculations in forward and backward mode.

*3) The authors on multiple occasions state that results are 'similar' (e.g. page 5 line 20, line 30). Can this 'similar' be quantified? Is it possible to use some statistical tests to assess whether the two experiments are statistically indistinguishable?*

In the manuscript we already have a table which shows for each model simulation pair (forward and backward) how many points fit within a certain percentage. Additionally we perform correlation analysis for each pair of forward and backward simulations, R is always above .89, which gives a significant correlation with p<0.001 in all cases. We also added this information in the text.

*4) Section 5 is at points particularly confusing. I is not clear why a MFB=-51% is considered accurate (page 8, line 18), while an MBF=-36% is considered an underestimation (page 8, line 30). Also this section would likely benefit from a map of these MFBs, as a new Figure perhaps*

As stated in the text (page 8 – line 26), the MFB can take values between -200 to +200. Therefore, both MFBs -51% and -66% show an underestimation of the model with respect to observations. Given the rather large uncertainties related to both modelling and observation of BC in snow, both values are actually not indicating a too bad agreement. We corrected a typo, it should be -66% rather than -36% and reformulated the respective sentences to be clearer now.
Concerning a map of FB, we created one and put it as S2 and added a reference to the Figure in the text.

*Furthermore, I also have some minor comments*
*- In the first paragraph of the introduction, this depends on the interpolation scheme used, I assume?*

We are not sure to which statement in the first paragraph your question refers to, so we cannot answer it. Sorry.

*- Page 2, line 34: calling ones own research a milestone is a bit grand.*

Indeed, this was an unfortunate formulation. We have changed this to:
... which would provide a very useful tool

*- Page 3, line 2: Remove 'briefly' here*

done.

*- page 3, line 13: How can both interpolation schemes be equally accurate? Do the authors mean 'exactly the same' here? Surely that can't be the case?*

This was perhaps not clear enough. The interpolation scheme is the same, but the coordinate systems are slightly different. We have rephrased this now to read:
"As the first meteorological input data set read into the model is different in forward and backward simulations, this can create small differences in the internal coordinate system used by the model, which can cause small differences in the interpolation of the meteorological data."

*- page 3, line 30: This sentence is quite confusing and vague for someone who isn't*

*experienced with using FLEXPART*
*We reformulated and gave more details:*

*"*The kernel assigns particle attributes to up to four grid cells, depending on the particle's position on the regular output grid (e.g., if a particle is located just at the boundary of two grid cells, both grid cells receive an equal fraction of the particle's attributes).*"*

*- page 4, line 8: how shallow is 'shallow' here?*

To explain this, we have added the following text in the manuscript:
"The height of this layer is equal to the height of the layer in which, in forward mode, particles are subject to dry deposition. By default, this height is 30 m, which is within the constant flux (surface) layer most of the time."

*- page 6, line 8-13: Are there any references from the literature here that can be used to back up these claims?*

Indeed, it is good to give references here. We have added "(see discussions on this topic in Seibert and Frank, 2004 and Lin et al., 2003)", as to our knowledge these are the only two papers discussing this.

*- page 7, line 7: refer to Figure 4 here?*

yes, we added a reference to Figure 4 to make it clear.

*- page 9, line 4: so the sources from Canada are definitely not anthropogenic?*

The fact that all other years except 2007 are captured well probably means that a specific source was active only in 2007. Given that anthropogenic emissions are more or less constant with time, we believe that the fluctuation is due to biomass burning emissions, which have been adopted from GFED. It has been shown in detail in the literature that changing certain parameters on how one calculates biomass burning from satellites, may lead to large discrepancies in the burned area calculation and hence in the final emitted mass. A very good example is Hao et al. doi:10.5194/gmd-9-4461-2016 (see Figure 2).

*- page 9, line 10: I am surprised there is no source at all from shipping (i.e. over the ocean). Is this a limitation of the technique? Or are there really no Black Carbon sources from shipping?*

The source of shipping in the emission inventory that was used to calculate BC in snow (ECLIPSEv5) is 0.6 Tg/year or 6.4% of the total annual emissions of BC from ECLIPSEv5 (~9.3 Tg/year). Furthermore, most shipping emissions occur far from where the samples were obtained (on Arctic sea ice and on land-based snow). While small contributions are present, they are, in fact, much smaller than their relative share of global emissions and are therefore not visible in our source contribution maps.

*- Figure 7: What is plotted on the background in this Figure?*

Both Figure 7 and 8 were created using the free Python's package Matplotlib. As the reviewer may see here:https://matplotlib.org/basemap/users/geography.html the background is NASA's bluemarble image. We added a reference to this in the figure caption

*- Figure 8left: is the unit here really in nanosecond per cubic meter?*

The units in Figure 8 were accidently set to the units of an air concentration ES. We correct this to the units m and now it is homogeneous through the manuscript.

*Type-os etc:*
*- Page 1, line 26 & 32: replace 'Firstly' and 'Secondly' with 'First' and 'Second'*
done.

*- page 5, line 18: 'extends over the entire atmospheric column'*

done.

*- page 6, line 6: 'with increasing number of particles'?*
correct. done.

*- page 7, line 21: Is 'nice' the right word here? Something more technical?*
We changed this word to "useful".

*- page 7, line 23: Is 'folding' a type-o?*
we meant multiplying, we replaced folding by multiplying.

---

## Author Comment (AC3) · 9 Oct 2017

[revised manuscript text omitted]

---

## Author Response (AR2)

Dear Editor,

thank you for handling my paper. I created a DOI at zendo as recommended and uploaded the version of FLEXPART I used in the paper. I tried to upload the dataset, but I failed. It is quite big. I have it here at my institute on a separated dedicated area and when someone asks for it I will provide the parts the person is interested in on our ftp.

Kind regards

Sabine Eckhardt